# Characteristics and Patterns of Proton Pump Inhibitors Prescribing at the Primary Health Care

**DOI:** 10.3390/medicina58111622

**Published:** 2022-11-10

**Authors:** Nataša Stojaković, Ana Golić Jelić, Svjetlana Stoisavljević Šatara, Nataša Bednarčuk, Miloš P. Stojiljković, Ranko Škrbić

**Affiliations:** 1Department of Pharmacology, Toxicology and Clinical Pharmacology, Faculty of Medicine, University of Banja Luka, 78000 Banja Luka, the Republic of Srpska, Bosnia and Herzegovina; 2Faculty of Medicine, University of Banja Luka, 78000 Banja Luka, the Republic of Srpska, Bosnia and Herzegovina; 3Academy of Sciences and Arts of the Republic of Srpska, 78000 Banja Luka, the Republic of Srpska, Bosnia and Herzegovina

**Keywords:** proton pump inhibitors (PPI), adverse drug reactions (ADR), the utilisation of medicines, physicians’ prescribing habits

## Abstract

*Background and objectives*: the aim of this study was to analyse the utilisation of proton pump inhibitors (PPIs) during a 12-year period and to show the characteristics and patterns of their prescribing. *Materials and methods*: firstly, in the pharmacoepidemiological analyses the ATC/DDD methodology was used to assess the utilisation of PPIs in the Republic of Srpska. The annual PPI utilisation was expressed as a number of DDD/1000 inhabitants/year. Secondly, the cross-sectional surveys were used to reveal the characteristics of PPIs prescribing and medicines use, namely the dose, duration and indication, and possible adverse reactions. For the purposes of the surveys, the adapted version of questionnaires related to physicians’ and patients’ perspectives of medicines prescribing and use were performed. *Results*: the utilisation of medicines for alimentary tract and metabolism (group A/ATC classification) increased by almost threefold in a 12-year period, which was consistent with the total medicine utilisation. Pantoprazole was the most prescribed medicine among the PPIs. With the exclusion of PPIs in the therapy of *Helicobacter pylori* eradication, more than half of family physicians prescribed PPIs with antibiotics, and only 53/239 physicians, noticed some adverse reactions of PPIs in their patients. Most of the patients knew how to use PPIs and were taking these medicines in recommended daily doses, but approximately 45% of them were using PPIs for a long period of time (>6 months). *Conclusions*: the overuse of PPIs is a major concern due to potential serious adverse reactions, especially in elderly patients and in a case of prolonged exposure.

## 1. Introduction

The use of proton pump inhibitors (PPIs) plays a key role in the treatment of peptic disorders such as dyspepsia, gastric or duodenal ulcer, gastro-oesophageal reflux disease (GERD), eradication of *Helicobacter pylori* infection and hypersecretory disorders, such as Zollinger Ellison syndrome [1,2]. Six randomised controlled clinical trials with 1259 participants assessed the effect of PPIs on the prevention of non-steroidal anti-inflammatory drugs (NSAID)—induced upper gastro-intestinal (GI) injury. PPIs significantly reduced the risk of both endoscopic duodenal ulcers and gastric ulcers, as compared with placebo [3,4]. PPIs are generally safe and effective medicines and since 2003 these medicines have been approved as ‘over the counter’ (OTC) products in many countries, providing patients with additional option for self-medication of dyspepsia and other gastric-related disorders [5,6]. However, the same factors have strongly contributed to their overuse and misuse. At the same time, physicians quite often prescribe PPIs for prolonged, even life-time use. Furthermore, many patients have been taking the OTC products beyond the recommended course of therapy without any supervision.

Over the years, there has been a growing concern about potential adverse reactions associated with long-term use of PPIs. Some of these are related to hypergastrinemia, development of pneumonia, dementia, drug-drug interactions and a variety of cardiovascular events [7,8,9,10]. Since 2010, the Food and Drug Administration (FDA) has issued several safety warnings regarding the potential side effects associated with long-term use of PPIs such as: osteoporosis and risk of fractures, hypomagnesaemia, Clostridium difficile-associated diarrhoea, vitamin B12 deficiency, acute interstitial nephritis and cutaneous and systemic lupus erythematosus [11,12,13,14].

The definitions of long-term PPIs treatment vary substantially among the studies [15]. Thus, in a clinical context, the use of PPIs for more than 8 weeks could be a reasonable definition of long-term use for patients with reflux symptoms and more than 4 weeks for patients with dyspepsia or peptic ulcer. For the purposes of pharmacoepidemiological studies, the use of PPIs for more than 6 months is often defined as long-term use, while studies of adverse reaction may require a tailored definition depending of the necessary exposure time [16]. 

The aim of this study was to analyse the use of PPIs during the period of 12 years and to show the characteristics and patterns of their utilisation using different perspectives and methods.

## 2. Methods

### 2.1. Pharmacoepidemiological Analyses

Retrospective, pharmacoepidemiological analyses were conducted to analyse the utilisation of PPIs in the Republic of Srpska during the period 2009–2020. The Republic of Srpska is one of two constitutive entities of Bosnia and Herzegovina (B&H) with a total population of 1.2 million, with executive and legislative functional responsibilities covering healthcare policies [17]. The Ministry of Health and Social Welfare is responsible for regulatory and executive functions, while the Health Insurance Fund (HIF) provides compulsory health insurance coverage for the entire population based on solidarity and mutuality [18]. Most of the prescribed medicines are reimbursed by HIF, and all prescribed medicines are dispensed in community pharmacies [19]. The national medicine utilisation data have been aggregated and published annually by the Public Health Institute (PHI) of the Republic of Srpska since 2009. The utilization of medicines was analysed using the Anatomical Therapeutic Chemical/Defined Daily Dose (ATC/DDD) methodology. Medicines were classified into ATC groups by their international non-proprietary names (INN). Data on outpatient medicines utilisation were expressed in DDD/thousand inhabitants per day (DDD/TID) for comparative purposes [20]. 

The share of group A in overall medicine use was calculated as Group A (DDD/TID)/Total medicine use (DDD/TID) × 100%.

The special focus of this analysis was put on the utilisation of PPIs, the ATC subgroup A02BC. These medicines were dispensed in community pharmacies as prescription medicines (omeprazole, pantoprazole, lansoprazole, esomeprazole, rabeprazole and dexlansoprazole) or as OTC medicines (pantoprazole, 20 mg 7 tablets, 20 mg 14 tablets).

### 2.2. Surveys

In order to acquire more valuable data related to characteristics, patterns and levels of PPIs prescribing at the primary health care level two specific cross-sectional surveys were conducted. The surveys were performed to reveal characteristics of PPIs prescribing and use, namely the dose, duration, treatment indications and possible adverse reactions. An adapted version of the questionnaire based on a previously published paper was used for data collection after obtaining official permission [21]. The original questionnaire was adapted by following the process of forward and backward translation and social-cultural adaptation [22]. Afterwards, two different questionnaires, one for family physicians and another one for patients who visited community pharmacies, were created and distributed from May to August 2021.

#### 2.2.1. Physicians’ Survey

After being adapted, the questionnaire for physicians was distributed to 239 family physicians from 11 primary healthcare centres, representing roughly 1/3 of all family physicians and covering the major geographic regions of the Republic of Srpska. The questionnaire included details of (1) physicians’ specialty, (2) duration of practice as a specialist, (3) medicines prescription characteristics (indication, dosage and duration of therapy), (4) reasons for proposed discontinuation of therapy, (5) the presence of adverse reactions associated with prolonged use of medicine and (6) physicians’ attitude related to prescribing appropriateness.

#### 2.2.2. Patients’ Survey

This part of the study was performed in collaboration with community pharmacists during the medicine dispensing process to patients. The questionnaire was distributed to 145 community pharmacies of the largest pharmacy chains in the Republic of Srpska. The management teams of pharmacy chains received a detailed information about the study design, interviewing methodology and the duration of data collection. In total, data from 812 patients who visited their pharmacies in 7-day period were collected. The questionnaire included details of the patients’: (1) age and sex, (2) INN and dose of prescribed medicine, (3) therapy dose regiment, (4) the origin of recommended therapy and reason for its use, and (5) suspected adverse medicine reactions.

### 2.3. Statistical Analyses

The results are presented as absolute numbers and percentages. All data were analysed using SPSS 20.0 (IBM Corp. Released 2011. IBM SPSS Statistics for Windows, Version 20.0 Armonk, NY, USA: IBM Corp.).

## 3. Results

### 3.1. Pharmacoepidemiological Analyses

During the 12-year period, the total medicines utilisation increased from 448.2 DDDs/TID in 2009 to 1258.1 DDDs/TID in 2020. At the same time the utilisation of medicines acting on the alimentary tract and metabolism, group A, increased by almost 3-folds, from 60.3 in 2009 to 183.7 in 2020. However, data share of group A medicines related to total medicines utilization showed a stable trend of consumption being between 11.4% and 14.6% (Table 1).

The utilisation of group A at the IV level of ATC classification showed a shift from H2 antagonists to PPIs (Figure 1). At the beginning of the observed period, the H2 antagonists were prescribed more frequently than PPI with the increase in the consumption of both groups of medicines until 2016 when they equalised. From 2018 to 2020 there was a significant increase in PPI consumption followed by a sharp decrease in consumption of H2 antagonists (Figure 1).

In 2020 the consumption of famotidine (H2 antagonist) increased slightly from, 0.6 in 2019 to 2.2 in 2020. 

At the beginning of the observed period, omeprazole was the most used PPI until 2012, then its use gradually decreased. During the same period, the utilisation of pantoprazole increased significantly, and it has become the leading PPI (Figure 2).

### 3.2. Surveys

#### 3.2.1. Physicians’ Survey

The data collected from the physicians’ questionnaires (*N =* 239) showed that most of the respondents were family physicians with more than 10 years in practice (Table 2).

Prescribing frequencies of PPIs for both groups were similar (based on their own decision or based on the decision of another specialist). Dyspepsia was the most common indication for PPIs prescribed by family physicians, while the most common indications for PPIs prescribing by other specialists were peptic ulcer, dyspepsia, and ulcer disease prophylaxis. Nearly half of the family physicians prescribed PPIs with antibiotics (excluded PPIs prescribed for the eradication of *Helicobacter pylori*), mainly with macrolides and tetracyclines (Table 3).

Since pantoprazole was the most prescribed PPI medicine (Figure 2), prescribing habits related to the dose regimen of this medicine were analysed. Most physicians prescribed pantoprazole in the following regimen: 20 mg once a day for dyspepsia, 40 mg twice a day for peptic ulcer, 40 mg once a day for the treatment of eradication of *Helicobacter pylori* infection and 20 mg once a day for the treatment of GERD. If prescribed for ulcer disease prophylaxis related to the use of other medicines (NSAIDs, corticosteroids, anticoagulants and antiplatelet medicines), most physicians prescribed pantoprazole in a dose of 20 mg daily (Table 4).

The most common reason for discontinuation of therapy was physicians’ adherence to guidelines in the treatment of certain diseases (Table 5).

The most recognised adverse drug reaction (ADR) due to PPI medicines was vitamin B12 deficiency (43.3% of all recognised ADR), followed by hypocalcaemia (26.8%), hypomagnesaemia (4.4%) and diarrhoea caused by Clostridium difficile (17.9%). None of physicians have ever identified ADRs related to pneumonia or acute tubulointerstitial nephritis. More than half (125/233) of the respondents had the opinion that medicines from the PPI group were over-prescribed.

#### 3.2.2. Patients’ Survey

As many as 812 patients were included in a survey that was conducted by pharmacists. Female and male respondents were equally represented, and most of them were from the age groups of 51–60 years and 61–70 years (Table 6).

Pantoprazole was the most used PPI medicine among the respondents (641/811) with the most frequent duration of therapy, up to 6 months, and the most used daily dose of 20 mg (526/620) that was taken before meals (757/809). 

The respondents started taking the PPI based on the recommendation of physicians (79.8%) or pharmacists (12.3%). Approximately 8% of respondents started taking PPIs on their own. Family physicians prescribed PPIs to 57% of the respondents. The initial dose regimen was adjusted/reduced in 87 patients, of which 71 adjustments were made by physicians and 16 by patients themselves.

The most common indications for taking PPIs were dyspepsia (233/786), prophylaxis of bleeding (219/786), and GERD (175/786) followed by the eradication of *Helicobacter pylori* (74/786), peptic ulcer (26/786), and other medical reasons (59/786).

Most of the patients reported that they never experienced any ADR due to PPI medicines, 97.8%. The respondents who confirmed the occurrence of ADR most often reported the following adverse reactions: nausea, diarrhoea, bloating and allergic reactions. The most frequently prescribed medicines together with PPIs were antiplatelet medicines followed by NSAID (Table 7).

## 4. Discussion

The pharmacoepidemiologic analyses clearly showed that the total medicine utilisation significantly increased almost 3-fold from 2009 to 2020 as well as the consumption of medicines from group A according to ATC classification (from 60.3 in 2009 to 183.6 in 2020). There was a growing trend of total medicines utilization over the time in the Republic of Srpska. The utilisation of group A medicines followed this trend with relative increase in share of 1%. At the same time we found a 6% increase in the relative share of cardiovascular (C group according to ATC classification) medicines utilization [23]. The reasons for these trends could be probably explained by better affordability and improved availability of medicines reimbursed by HIF. Medicine utilisation of group A at the IV level of ATC classification showed a clear shift from H2 antagonists to PPIs. A significant increase in PPIs consumption was noticed in 2018, followed by a sharp decrease in consumption of H2 antagonists. The decrease in H2 antagonist utilisation could be ascribed to the withdrawal of ranitidine from the market in October 2019 due to the presence of increased levels of impurities [24]. Since that time, only famotidine, as H2 antagonist was registered in our country, but that medicine was not reimbursed by HIF. Thus, in the PPI class of medicines there were significantly more registered INNs, including pantoprazole as both an OTC and Rx medicine, depending on the dose and package. Over the years, ranitidine had never been replaced with famotidine but was replaced almost exclusively by PPIs. There was a slight increase in the consumption of famotidine in 2020 due to the efforts of local pharmaceutical producers to compensate the withdrawal of ranitidine from the market, but in comparison with the consumption of PPIs, it was negligible. In 2020 the COVID-19 pandemic seems not to have influenced the total utilization of drugs, which can be explained with by effective control measures and public health interventions [25].

Such increase in PPI consumption (from 2.8 to 23.4 DDD/TID between 2009 and 2020) is in accordance with data from Hungary and Nordic countries. In Hungary, the PPIs utilisation increased from 41.9 to 50.4 DDD/TID between 2014 and 2018 [21]. In Nordic region countries, H2 antagonists’ consumption decreased during the period from 2005 to 2016, while PPIs consumption increased more than 2-fold, which is similar to our results [26]. The same trend with steady increase in PPIs utilisation was noticed in neighbouring countries such as Serbia and Croatia [27,28].

The data from the physicians’ survey showed that most of the respondents were family physicians with more than 10 years in practice, meaning that they are experienced enough and eager to follow new trends in disease management. Physicians prescribed PPI medicines at similar frequencies on their own decision as well as following the recommendation of another specialist. Prescriptions based on their own decision were mostly for dyspepsia, while the most prescribed PPIs due to other specialists’ recommendations were for peptic ulcer, dyspepsia, and ulcer disease prophylaxis. However, with the exclusion of PPIs in the therapy of *Helicobacter pylori* eradication, it is a worrying finding that nearly half of the physicians regularly prescribed PPIs with antibiotics (mostly macrolides and tetracyclines).

Dyspepsia was the most common reason for pantoprazole prescribing among family physicians, which is consistent with the most used dose of pantoprazole (20 mg a day) for this indication, while peptic ulcer, dyspepsia and ulcer disease prophylaxis were the most common reasons for pantoprazole prescribing among other specialists. The same dose of pantoprazole was prescribed for prophylaxis of ulcer disease due to the use of other medicines such as NSAIDs, corticosteroids, anticoagulants, or antiplatelet medicines, what is in accordance with guidelines for prolonged use of PPI for this indication [29]. 

Approximately 46% of patients from our study have used PPI for more than 6 months. Recent studies suggested that prolonged PPI exposure was potentially associated with serious adverse reactions including chronic kidney diseases (CKD), cardiovascular diseases and gastrointestinal malignancies, with a small excess of cause-specific mortality [29,30]. The incidence of ADRs increases not only with prolonged use of PPIs but also with increasing doses. Rebound phenomenon is not associated with other medicines used for dyspepsia, such as antacids, alginates and H2 antagonists, as it happens after discontinuation of PPI therapy. 

In the present study, the most common reason for discontinuation of therapy was physicians’ readiness to follow guidelines in the treatment of certain diseases. The need for PPI therapy should be assessed regularly to reduce its high consumption, having in mind that inappropriate use of PPIs is a matter of great concern, especially in the elderly [31,32].

We noticed that only 53 out of 239 (%) physicians stated that they noticed some ADRs of PPIs in patients, which indicates the need for further education of physicians. None of the physicians identified ADRs related to pneumonia and acute tubulointerstitial nephritis. It is possible that PPI use may also be a risk factor for CKD, potentially mediated by recurrent acute kidney injury or by hypomagnesaemia, which has been associated with PPIs use and with incidence of CKD [33].

Our results showed discrepancies when compared answers from physicians and patients. Great number of patients (around 45%) were using PPIs for a long period of time, more than 6 months. The guidelines for most gastrointestinal diseases recommend that PPIs should not be prescribed for more than 6 months. However, the widespread use of PPIs in the Republic of Srpska seems that its utilisation is not in accordance with the guidelines. In addition, for most of the patients (89.28%) the dosage regimen had never been adjusted, which the opposite of the physicians’ statements. Data related to dispensed medicines together with PPIs during the same patient visit to pharmacy indicate that most of PPIs are still prescribed for prophylaxis of ulcer disease due to concomitant use of other medicines such as: NSAID, antiplatelet medicines and corticosteroids. A similar situation has been observed in other countries where the misuse of PPIs for gastroprotection related to NSAID therapy was also observed both, regarding underuse in at-risk patients and overuse in non-at-risk patients. For 53.5% of new users of PPIs in a clinical study conducted in France, indication for PPIs therapy was a co-prescription with NSAIDs; in this indication, most of the patients (79.7%) had no measurable risk factor supporting a systematic prophylactic co-prescription of PPIs. In the same study, a proportion of 32.4% of new users did not have any identified co-medication or inpatient diagnosis supporting an indication for PPI therapy [31].

It is interesting that in questions related to ADRs of PPIs, the great majority of patients (97.7%) generally answered negatively, which is consistent with replies from physicians who noticed some ADRs related to PPIs. According to the available data, the incidence of ADR for PPIs is higher than reported by other authors, which is a matter of concern keeping in mind that neither physicians nor patients recognize the possible danger of ADRs [8,9,34]. These results clearly emphasize the need for additional and continuous education of physicians, pharmacists, and patients related to possible ADRs, as it was shown previously [18,35]. 

### Strengths and Limitations

The strengths of this study are related to overall outpatient utilization of medicines in the Republic of Srpska based on data obtained from PHI comprising all medicines including not only reimbursed ones, but also the OTC medicines as well. Additionally, the surveys combined different perspectives of PPIs utilization from prescribers’ to patients’ perspective. 

However, the present study has some limitations. Firstly, this study did not include precise data on hospital utilization of PPIs which could potentially influence the overall consumption of this medicines. Additionally, some information related to indication, initiation, and duration of therapy, as well as prescribing were based on patients’ statements that could be subject of recall bias.

## 5. Conclusions

During the 12-year observed period the utilisation of medicines acting on the alimentary tract and metabolism increased by almost 3-fold. Pantoprazole was the most commonly prescribed PPI since significant increases in PPIs utilisation were registered from 2018 until today. Nearly half of the family physicians prescribed PPIs with antibiotics for indications that did not include eradication of *Helicobacter pylori*. In addition, the most frequently prescribed medicines together with PPIs were antiplatelet medicines, followed by NSAID. Our study identified a low detection rate of ADRs related to PPIs, both by physicians and by patients compared to their incidence in the world. Approximately 45% of patients were using PPIs for a long period of time, with the initial dose regimen adjusted/reduced on physician’s recommendation in only 8.7% of patients. The PPI overuse is a major concern due to unrecognized, potentially serious adverse events, especially in case of prolonged exposure.

## Figures and Tables

**Figure 1 medicina-58-01622-f001:**
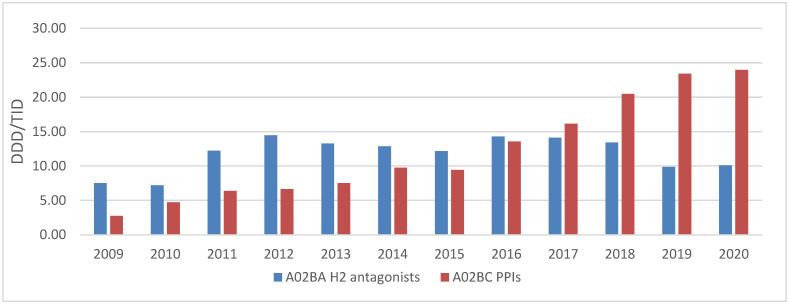
The utilisation of medicines from A02BHA (H2 antagonists) and A02BC (PPIs) groups at the IV level according to ATC classification, expressed in DDD/TID.

**Figure 2 medicina-58-01622-f002:**
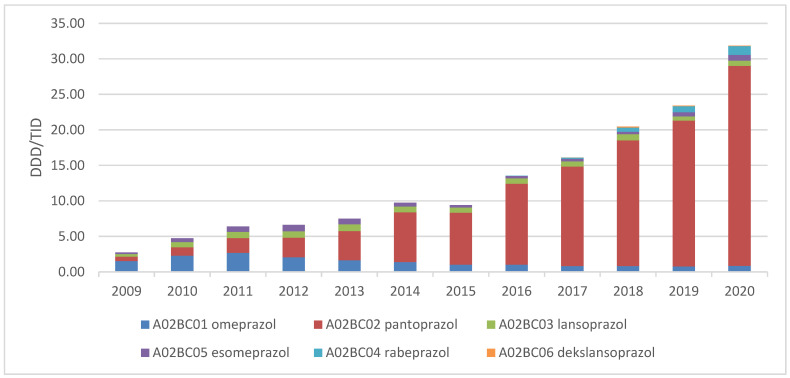
The utilization of proton pump inhibitors expressed in DDD/TID during the period 2009–2020.

**Table 1 medicina-58-01622-t001:** The utilisation of medicines acting on the alimentary tract and metabolism (group A according to ATC classification) expressed in DDD/TID and as a share (%) of the total medicines’ utilisation in the Republic of Srpska during the period 2009–2020.

Year	2009	2010	2011	2012	2013	2014	2015	2016	2017	2018	2019	2020
Total medicine use in DDD */TID **	448.2	622.3	746.6	731.6	764.3	841.7	861.2	969.5	1036.3	1225.9	1158.0	1258.1
Group A DDD/TID	60.3	70.8	105.6	105.3	106.7	110.4	108.1	123.1	126.6	145.2	144.7	183.7
Share *** (%)	13.5	11.4	14.1	14.4	14.0	13.1	12.6	12.7	12.2	11.8	12.5	14.6

* DDD/Defined Daily Doses; ** TID—Thousand Inhabitants per Day; *** Share of group A in overall medicine use.

**Table 2 medicina-58-01622-t002:** Sociodemographic and work characteristics of respondents, group of physicians (*N* = 239).

Sociodemographic and Work Characteristics		*N*	%
Family physician			
	Yes	157	65.7
	No	82	34.3
Length of specialist status			
	Less than 5 years	18	7.5
	5 to 10 years	31	13.0
	More than 10 years	108	45.2

**Table 3 medicina-58-01622-t003:** Physicians prescribing habits of PPIs (*N* = 239).

Physicians Prescribing Habits		*N*	%
Frequency of prescribing based on their own decision			
	Very often	70	29.3
	Often	139	58.2
	Rarely	28	11.7
	I do not prescribe	2	0.8
Frequency of prescribing based on the decision of another specialist			
	Very often	57	23.8
	Often	151	63.2
	Rarely	27	11.3
	I do not prescribe	4	1.7
Most frequent indication (based on their own decision)			
	Dyspepsia	114	47.7
	Eradication of *Helicobacter pylori*	52	21.8
	Prophylaxis of ulcer disease due to the use of other medicines	33	13.8
	Peptic ulcer	24	10.1
	GERD	14	5.9
Most frequent indication (based on the decision of another specialist)			
	Dyspepsia	59	24.7
	Eradication of *Helicobacter pylori*	26	10.9
	Prophylaxis of ulcer disease due to the use of other medicines	57	23.9
	Peptic ulcer	57	23.9
	GERD *	32	13.4
Prescribing PPI with antibiotics (excluded PPIs prescribed in the eradication of *Helicobacter pylori*)			
	Yes	106	44.4
	No	133	55.7
	Cephalosporin	32	13.4
	Macrolide	72	30.1
	Tetracycline	53	22.2
	Penicillin	25	10.5
	Fluor quinolone	39	16.3

* GERD—Gastroesophageal reflux disease.

**Table 4 medicina-58-01622-t004:** Physicians’ prescribing habits of pantoprazole (*N* = 239).

Indication	Dose Regimen	*N*	%
Dyspepsia			
	20 mg once a day	146	61.1
	20 mg twice a day	12	5.0
	40 mg once a day	69	28.9
	40 mg twice a day	1	0.4
Eradication of *Helicobacter pylori*			
	20 mg once a day	25	10.5
	20 mg twice a day	18	7.5
	40 mg once a day	103	43.1
	40 mg twice a day	85	35.6
Prophylaxis of ulcer disease due to the use of other medicines			
	20 mg once a day	173	72.4
	20 mg twice a day	22	9.2
	40 mg once a day	31	13.0
	40 mg twice a day	4	1.7
Peptic ulcer			
	20 mg once a day	22	9.2
	20 mg twice a day	24	10.0
	40 mg once a day	55	23.0
	40 mg twice a day	127	53.1
GERD			
	20 mg once a day	92	38.5
	20 mg twice a day	30	12.6
	40 mg once a day	74	31.0
	40 mg twice a day	37	15.5

**Table 5 medicina-58-01622-t005:** Physicians’ reasons for discontinuation of PPI therapy (*N* = 135).

Reasons for Discontinuation of Therapy	*N*	%
Adverse drug reaction	49	22.3
Inadequate therapeutic response	16	7.8
Adherence to guidelines in the treatment of the disease	70	31.8

**Table 6 medicina-58-01622-t006:** Patients’ socio-demographic and PPIs utilisation characteristics, *N* = 812.

Variables		*N*	%
Age (years)			
	0–10	0	0
	11–20	2	0.2
	21–30	42	5.2
	31–40	106	13.1
	41–50	130	16.0
	51–60	166	20.5
	61–70	201	24.8
	71–80	132	16.3
	81–90	31	3.8
	91–100	1	0.1
Prescribed/dispensed PPI medicine			
	pantoprazole	641	79.0
	omeprazole	41	5.1
	esomeprazole	35	4.3
	lansoprazole	23	2.8
	rabeprazole	76	9.4
Duration of PPI therapy			
	<1 month	229	28.2
	2–6 months	160	19.7
	7–12 months	122	15.0
	1.5–3 years	137	16.9
	>3 years	114	14.1
	Optionally	35	4.3
	A long time/indefinitely	13	1.6
Adjustment of initial dose regimen			
	Yes	87	10.7
	No	724	89.3

**Table 7 medicina-58-01622-t007:** The most dispensed medicines together with PPIs.

Medicines	*N*
Antibiotics as a part of triple therapy	60
Antibiotics	12
NSAID	73
Antiplatelet medicines	128
Corticosteroids	9

## Data Availability

Not applicable.

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
