# Peer review of "Characteristics and Patterns of Proton Pump Inhibitors Prescribing at the Primary Health Care"

_medicina, 2022, doi:10.3390/medicina58111622_

Round 1
Reviewer 1 Report
This paper is based on a survey of doctors and patients and an analysis of the use of PPI in the country in the past 12 years. These results are very meaningful and are of great value in understanding doctors' preferences for using different PPI, patients' awareness of PPI, and the formulation of health insurance policies.
Author Response
Respected reviewer,
thank you for taking the time to review our manuscript.
Kind regards
Reviewer 2 Report
The manuscript titled "Characteristics and patterns of proton pump inhibitors prescribing at the primary health care" aimed at analysing the pattern of utilization of drugs for alimentary tract and metabolism, including proton pump inhibitors, during a 12-year period in the primary care setting of the Republic of Srpska. It also provides characterists and perspectives of physicians and patients using these medicines.
I would like to provide some comments for your consideration in order to increase the overall quality of the research paper.
INTRODUCTION
-Please consider to mention the risk of cardiovascular events as potential adverse drug reaction associated with PPI use (even though it is still subject of debate)
METHODS
-How many PPI users were included in the study for each year? How many physicians from primary care? Please provide a sample size estimate for the pharmacoepidemiological analysis
-What age groups (adolescents? adults? elderly?) were analysed? In my opinion, it is important to distinguish between different age groups, especially considering that the prescription of proton pump inhibitors for more than 8 weeks in persons aged ≥ 65 years is included in the list of PIM (potentially inappropriate medications) for older people
-DID is commonly used to express DDD/1000 inhabitants per day. What does DIDs/TID (line 83) refer to?
-Please add how to calculate the “Share” of group A in overall medicine use reported in Table 1
-Please number the paragraphs and sub-paragraphs (2.1, 2.2, etc.) as done in the Results section
RESULTS
-Is there a reason for such a growing trend of total medicines utilization over time in the Republic of Srpska? On the other hand, how do the authors comment on the stable trend showed by the “Share” of group A in overall medicine use as reported in Table 1?
-Also, in 2020 the COVID-19 pandemic seems not to have influenced the general utilization of drugs. How do the authors comment on this point?
-Line 126: The value for 2009 (63.2 DID) is different from the one reported in table (60.3). Please review
-Line 129: Please complete “Group A according”
-Line 164: Please review the statement “More than half of the family physicians prescribed PPIs with antibiotics”. Indeed, it is in contrast with what reported in Table 3 (Yes à 44.4%). The same statement is repeated in the discussion and conclusion sections.
DISCUSSION
- Line 230: What IPPs stand for?
- Line 260-261 needs rewording
- Line 294-295 needs attention
- Description of strengths and limitations of these studies are not provided (both pharmacoepidemilogical analyses and surveys)
CONCLUSIONS
- Line 310-312: Statement not directly related to the results of this study. Please review
REFERENCES
- The format of the references does not meet the journal guidelines
Author Response
Respected reviewer,
thank you for taking the time to review our manuscript. Your suggestions and comments were very valuable and revised version of manuscript is now improved.
We attached the file with detailed response for every point you stated. Please see attachment.
Kind regards
